# Enhancing the Generalization for Text Classification through Fusion of Backward Features

**DOI:** 10.3390/s23031287

**Published:** 2023-01-23

**Authors:** Dewen Seng, Xin Wu

**Affiliations:** School of Computer Science and Technology, Hangzhou Dianzi University, Hangzhou 310005, China

**Keywords:** deep learning, text classification, two-stream networks, feature fusion, sentiment classification, sarcasm detection

## Abstract

Generalization has always been a keyword in deep learning. Pretrained models and domain adaptation technology have received widespread attention in solving the problem of generalization. They are all focused on finding features in data to improve the generalization ability and to prevent overfitting. Although they have achieved good results in various tasks, those models are unstable when classifying a sentence whose label is positive but still contains negative phrases. In this article, we analyzed the attention heat map of the benchmarks and found that previous models pay more attention to the phrase rather than to the semantic information of the whole sentence. Moreover, we proposed a method to scatter the attention away from opposite sentiment words to avoid a one-sided judgment. We designed a two-stream network and stacked the gradient reversal layer and feature projection layer within the auxiliary network. The gradient reversal layer can reverse the gradient of features in the training stage so that the parameters are optimized following the reversed gradient in the backpropagation stage. We utilized an auxiliary network to extract the backward features and then fed them into the main network to merge them with normal features extracted by the main network. We applied this method to the three baselines of TextCNN, BERT, and RoBERTa using sentiment analysis and sarcasm detection datasets. The results show that our method can improve the sentiment analysis datasets by 0.5% and the sarcasm detection datasets by 2.1%.

## 1. Introduction

Text classification is an essential and vital branch of the natural language process (NLP). It has received widespread attention from many scholars who utilize neural networks to extract high-quality semantic features from inputs such as sentences and documents. The most classic model among the neural network models is the convolutional neural network [1]. This model can extract highly representative semantic features for classification. Although the CNN-based model can effectively capture local- and fixed-position features, its accuracy still needs to be improved. The most recent model is BERT, proposed by Kenton et al. [2]. BERT and RoBERTa [3] use pretrained technology on large datasets to capture the universal information to improve the generalization ability.

However, even if these algorithms could achieve a state-of-the-art performance, there would still be room for improvement. From the perspective of features, these algorithms make mistakes in specific sentences. For example, when the trained RoBERTa judges the sentence, a “charming and funny (but ultimately silly) movie”, RoBERTa classifies it as a negative comment. However, it is a positive comment. We analyzed the attention heat map and can assume that RoBERTa pays more attention to the phrase “but ultimately silly” rather than “charming and funny”, which is the reason for the mistake. The heat map of RoBERTa’s analysis of this sentence is shown in Figure 6.

To address this point, our study follows the research ideas and processes presented in this paper to allow models to be able to scatter the attention of opposite sentiment words given to core sentiment words. In this paper, we utilize a two-stream network structure to further study the relative differences between the backward features of the auxiliary network and normal features of the main network to improve the representation ability of feature vectors. For example, we first extract backward features through an auxiliary network and then use a feature projection layer to obtain the extra vector, which has the same direction as the normal feature. After that, to leverage the balance between normal feature vectors and extra vectors, we study a method to aggregate normal features and extra vectors that are projected after the projection layer. We assume that such a design has the potential to analyze more information about the context through end-to-end training.

Inspired by the feature purification network (FP-Net) [4], we propose a method called the feature augmentation network (FA-Net). Separately, F-Net refers to the main network, and A-Net refers to the auxiliary network. The A-Net uses the gradient reversal layer [5] to extract the backward features that contain backward contextual information. Meanwhile, the F-Net is a normal network such as a CNN or BERT. This means the major work carried out by the F-Net is meant to extract normal feature vectors. Before feeding the normal vector into the classifier, we calculate an extra vector through feature projection that is in the same direction as the normal vectors. After that, we concatenate normal vectors and extra vectors together, creating a new feature vector. Finally, the model feeds a new vector into the classifier. This study makes three main contributions:The parameters of our model are acceptable. Even in the BERT-based model, our model only has one or two additional encoder layers.Our algorithm is efficiently utilized at different benchmarks, such as with the CNN and BERT, and it is not conflicted with other operations that improve generalization capabilities.We analyze the influence of the auxiliary network on the attention score of the main network, expressing the efficiency of the auxiliary network through the attention heat map.

To better explain the proposed methods, we introduce relevant research on text classification and feature fusion and briefly describe their practices in Section 2. In Section 3, we describe the six open-source datasets and three open-source models that are used as the experiment materials. We also focus on introducing our model’s structure in Section 3. Later, we list our experimental data in Section 4. To show the effectiveness of our method, we implement it on sentiment analysis and sarcasm detection datasets. We list the average results under five seeds and illustrate the stability of the model through deviation. We also analyze our experimental results and discuss the projection type, the number of subnetworks, and the type of subnetworks. We also prove that our idea is consistent with the hypothesis from the perspective of the attention heat map. In Section 5, we explain our conclusions and future prospects.

## 2. Related Works

The well-known RNN model used for text classification is long short-term memory (LSTM) [6,7]. LSTM uses the forget gate to choose whether to retain the previous information or not. Thus, LSTM is good at processing long-term-dependent input. However, compared to CNNs, LSTM does not run fast, causing some scholars to turn their attention to the CNN models, which can operate fast and parallel to the training stage.

The TextCNN [1] sets fixed filter sizes that work on embedded vectors to capture context information. Then, the maxpooling focus is used on salience features. To obtain more information on these features, Wang et al. [8] proposed a method that uses a concentration mechanism to pick out the key features for short text classification. However, the problem with TextCNN is that it is hard to obtain the long-term information because of the n-gram mechanism of convolution filters, which can only operate on several consecutive words simultaneously. Therefore, Lai et al. [9] combined the advantages of the TextCNN and RNN and designed the TextRCNN algorithm. They utilized the Bi-RNN to build the left and right contexts, then concatenated those vectors with embedding vectors to feed them into one neural layer and generate the latent semantic vector and finally extracted the feature through max pooling, as is performed via the TextCNN. After the TextCNN, many advent approaches related to neural networks were proposed, such as, e.g., the DCNN [10], HAN [11]. The DCNN uses dynamic k-max pooling to extract long-distance features that were separated dynamically. The HAN uses a multilayer recurrent neural network and an attention mechanism to extract long-sentence semantic features.

Although some people are still studying the RNN and CNN model structures for text classification, the naive attention [12] mechanism was proposed. After this, many variants of attention mechanisms appeared, such as local attention, global attention, and soft attention. The most effective attention mechanism is self-attention. As proposed by Vaswani et al. [13], this model effectively reduces the calculation cost by parallel computing the attention score of each word in the text or document. Furthermore, based on transformer and attention mechanisms, some scholars utilized other fields’ technologies to improve the model’s performance. BERT combines pretrained technology, transformer encoders, and the training of a vast corpus to extract more comprehensive feature vectors and achieve an SOTA performance for a wide range of tasks. Based on BERT, some scholars proposed more effective models, such as RoBERTa. This method changed static masking to the dynamic masking of sentences and removed next-sentence prediction (NSP); thus, its main difference with BERT is in the pretraining stage. Indeed, this way improves the performance of BERT.

Due to the highly expressive ability of BERT’s encoder, many scholars take the features extracted by the encoder as their research focus. Qin et al. [4] proposed a feature projection layer to eliminate the redundant information of features and improve the quality of features. G Niu et al. [14] proposed a new Encoder1–Encoder2 structure, where Encoder1 is a global information extractor and Encoder2 is a local information extractor. The global information vectors are merged with the local information vectors for a higher performance. Ying et al. [15] proposed an unsupervised saliency detection approach, which utilizes an elastic-net-based hypergraph model to discover the group structure relationships of salient regional points. They also use a saliency map to obtain high-level semantic features. Then, they fused the low-level deep and high-level semantic features into a similarity matrix. Wang et al. [16] proposed a novel structure comprising three modules. One of the modules is responsible for multiscale feature alignment fusion. The other modules are focused on different scale channels and the adaptive weighted fusion of spatial locations, as well as the multiscale fusion of global and local features. Long et al. [17] proposed a method for mining the relationships between labeled and unlabeled data. They used the co-occurrence of words in all documents to build a neighbor table and use multidimensional scaling (MDS) to extract the feature representation of the adjacency table. Then, they integrated the new graph-based representation and the document–term representation as the new hybrid augmented feature representation. Huang et al. [18] added other cheap modules, called Ghost Modules, to capture more semantic information and then fused them with normal features extracted by the base model. They studied the 2D convolutional operation and 1D convolutional operation of the Ghost Modules. Additionally, they also analyzed which position was better when inserting their Ghost Modules within the transformer encoder.

## 3. Materials and Methods

### 3.1. Methods

In this paper, we mainly studied the deep learning approach to improve the quality of feature vectors. Our method has a built-in feature projection layer and gradient reversal layer in the auxiliary network. For the projection layer, in our implementation, we decomposed the backward features in two directions and chose the one that has the same direction as the normal feature, keeping in line with our assumption. Another direction is not suitable for our method because the effectiveness is not apparent. The gradient reversal layer is vital to help the model scatter the attention score of a phrase whose semantic meaning is opposite to that of the sentence. Because the gradient reversal layer can reverse the gradient of backward features, a normal feature fused with the extra feature generated by the projection layer can contain the reversed gradient information to force the model to focus on core words other than the contradictory words. We conducted our experiments under the different types and sizes of the subnetwork to show the different results. For example, we cut down the number of filter sizes in the CNN model to find an appropriate and time-effective extractor for the auxiliary network. Additionally, we did not use the convolution layer to extract extra features except in the CNN models. In contrast, we utilized the transformer encoder to extract additional features, as it can generate more explainable features than the convolution layer. The overview of the structure of our FA-Net is described in Figure 1.

As shown in Figure 1, our network consists of two networks. The F-Net attentively extracts the normal features $v_{f}$ by utilizing the extractor $E_{f}$ with perturbation from the A-Net. On the other side, the A-Net is focused on extracting backward features $v_{a}$ through the gradient reversal layer. In our implementations, the type of $E_{a}$ is the same as that of $E_{f}$, which means the layer type of the A-Net follows that of the F-Net. If the F-Net utilizes the convolutional layer to extract the features, the A-Net does the same. The other exciting settings are the size and inputs of $E_{a}$. The size of extractor $E_{a}$ is smaller than the size of $E_{f}$. The input of $E_{a}$ is the cloned output from one of the encoders of $E_{f}$. After feeding the cloned feature vector into the A-Net, the output of the A-Net is entered into the projection layer with the output of the F-Net. The final feature vector is obtained through the projection layer and fed into the classification layer to generate the output of the whole network.

In the backpropagation stage, we initialized two optimizers. The optimizer of the F-Net is responsible for updating the F-Net’s parameters and the parameter of the embedding layer, and the optimizer of the A-Net is accountable for updating the A-Net’s parameters. To explain the algorithms directly, we list the algorithm procedure in Algorithm 1.

**Algorithm 1: **Feature Augmentation Network

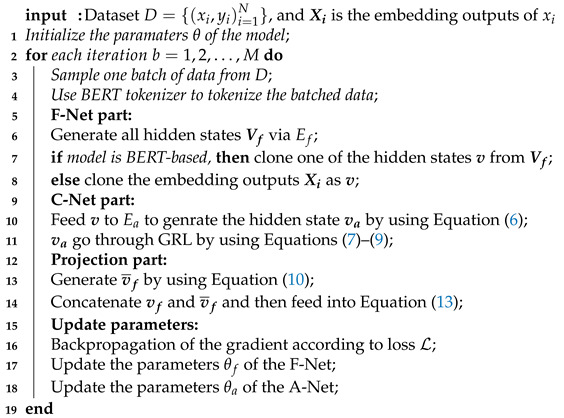



Although our network consists of two networks, both networks share one loss function L. The advantage of this is that the amount of time consumed by the model is reduced. The loss function of the whole network is as follows:(1)$$\begin{array}{c} L=L_{f}=L_{a} \end{array}$$

We introduced the proposed method by following the structure of the FA-TextCNN as an example. The FA-TextCNN is a model that applies our method to the TextCNN. Each part is as follows:

F-Net Module: For the FA-TextCNN, a dataset $D=\{\left(x_{i},y_{i}\right)|i\in 1,\ldots ,N\}$ is given, where $x_{i}$ is a sentence or document with the corpus length *L* (after padding or cutting), *N* is the size of the training data, and $y_{i}$ is the label of $x_{i}$. Here, $x_{i}$ feeds into the embedding layer with a fixed embedding size *e* to generate the embedded output $X_{i}\in R^{L\times e}$. Whereafter, $X_{i}$ feeds into the feature extractor $E_{f}$ with convolutional filters and n-gram to generate features $v_{f}$ as follows:(2)$$\begin{array}{c} c_{i}^{j}=f\left(W\cdot X_{i}\left[j:j+n-1,:\right]+b\right) \end{array}$$
(3)$$\begin{array}{c} c_{i}=\left[c_{i}^{0},c_{i}^{1},\ldots ,c_{i}^{L-n}\right] \end{array}$$
where j∈0,…,L−n and $W\in R^{n\times e}$ is the weight of the convolution filter, and *n* is the n-gram size of each convolutional filter. Moreover, *f* is the active function, similar to *ReLU*. The outcome of feature $f_{c}$ under the n-gram and a filter is as follows:(4)$$\begin{array}{c} f_{c}=\left[c_{0},c_{1},c_{2},\ldots ,c_{L-n}\right] \end{array}$$

After this, we used the maxpooling operation over the feature map and took a maximum value $m_{f}=max\left\{f_{c}\right\}$ as the most characteristic feature under the one filter. In our experiments, we initialized *q* filters, and each filter initialized *m* kinds of parameters. Therefore, a filter can generate one of the most characteristic values of a parameter. Finally, we concatenated those characteristic values. Obviously, the $v_{f}\in R^{q\cdot m}$ is extracted by $E_{f}$ as follows:(5)$$\begin{array}{c} v_{f}=CNN_{f}\left(X_{i}\right) \end{array}$$

A-Net module: This module is our designed network. The output of the embedding layer $X_{i}$ is cloned, and then $X_{i}$ is fed into $E_{a}$, which is similar to $E_{f}$. Because the $E_{a}$ is also a convolution filter that can set the n-gram and kinds of parameters, it can generate the feature $v_{a}$ of $X_{i}$ under a specific convolution filter:(6)$$\begin{array}{c} v_{a}=CNN_{a}\left(X_{i}\right) \end{array}$$

Upon $E_{a}$, we innovatively stacked the GRL on the extractors of the A-Net and used the projection layer to eliminate the harmful semantic information of backward features $v_{a}$. The procedure of the gradient reversal layer is as follows:(7)$$\begin{array}{c} GRL_{\lambda }\left(x\right)=\tilde{x} \end{array}$$
(8)$$\begin{array}{c} \frac{\partial GRL}{\partial x}=-\lambda I \end{array}$$
where λ is a hyperparameter of the gradient reversal layer, and $\tilde{x}$ is a new feature vector passed through the gradient reversal layer. We noted that the classification of our whole model is mainly completed through the F-Net. Therefore, to reduce the influence of the A-Net at the beginning of training, we gradually increased the λ as follows:(9)$$\begin{array}{c} \lambda =\frac{2}{1+exp(-\gamma \cdot p)}-1 \end{array}$$
where γ was set to 10 in all experiments, and *p* represents the iteration ratio of training from 0 to 1.

When ready, $v_{f}$ and $v_{a}$ were both fed into the projection layer to generate ${\bar{v}}_{f}$ and ${\bar{v}}_{a}$:(10)$$\begin{array}{c} {\bar{v}}_{f}=Proj\left(v_{a},v_{f}\right) \end{array}$$
(11)$$\begin{array}{c} {\bar{v}}_{a}=Proj\left(v_{f},v_{a}\right) \end{array}$$
where ${\bar{v}}_{f}$ is the projected feature suited for us, and ${\bar{v}}_{a}$ is the tested feature that we concatenated with the normal feature, as is explained in the Discussion section of this paper. Additionally, *Proj* is a projection function that projects a vector to another:(12)$$\begin{array}{c} Proj\left(v_{x},v_{y}\right)=\frac{v_{x}\cdot v_{y}}{|v_{y}{|}^{2}}\cdot v_{y} \end{array}$$

After the projection layer, we concatenated two features, $v_{f}$ and ${\bar{v}}_{f}$, as a new vector $v_{f}^{*}$, then fed $v_{f}^{*}$ into the classifier. Finally, we utilized the *Softmax* function to achieve the classification and used the *CrossEntropy* function as our *Loss* function:(13)$$\begin{array}{c} v_{f}^{*}=concat\left(v_{f},{\bar{v}}_{f}\right) \end{array}$$
(14)$$\begin{array}{c} Y_{f}=Softmax\left(v_{f}^{*}\cdot W_{f}+b_{f}\right) \end{array}$$
(15)$$\begin{array}{c} Loss_{f}=CrossEntropy\left(Y_{truth},Y_{f}\right) \end{array}$$
where $W_{f}\in R^{q\cdot m⊕q_{a}\cdot m_{a}\times C}$, $q_{a}$ is the number of filters of $E_{a}$, $m_{a}$ is the kind of initialized filter for each filter, and $b_{f}\in R^{C}$, *C* is the number of labels. $Y_{truth}$ is the marked label, and $Y_{f}$ is the label predicted by the F-Net. The entire model that is used in the experiments on the FA-TextCNN is shown in Figure 2. Furthermore, we used a two-dimensional vector projection process to more intuitively express what the feature projection layer does in Figure 3.

Keep in mind that we implemented our method not only on CNN-based, but also on BERT-based models. To express our idea more clearly, we also drew a figure of the FA-Net being used on the BERT-based model, as shown in Figure 4. In the BERT-based model, some details do not align with the TextCNN. We weighed the time consumption and accuracy of the FA-Net by referred to the analysis of Sun et al. [19]. Consequently, we identified the high-level encoder from the base BERT as the encoder of $E_{a}$ because its layers’ output had a good classification ability. We noticed that the gradient of the A-Net is climbed up and then went back to the F-Net, influencing the parameters of the F-Net further. We decided to feed different kinds of the output of extractor $E_{f}$ to extractor $E_{a}$ to study the different types of projection operations. Therefore, we chose different n-gram sizes in the FA-TextCNN model and different encoder layers in the FA-BERT and FA-RoBERTa models. One reason we tried different inputs of $E_{a}$ is that there are two projection types. For each type, we tried the same input to verify which is better. Another reason is that the input of $E_{a}$ is cloned from the F-Net. Naturally, the A-Net gradient can go back to the F-Net. The higher the number of encoder outputs picked up by $E_{a}$, the more the encoders of the F-Net are influenced by the gradient of the A-Net.

On the one hand, we want to feed high-quality feature vectors to $E_{a}$. On the other hand, we want the gradient of the A-Net to affect the F-Net as little as possible. Thus, experiments on different inputs of $E_{a}$ are necessary.

**Figure 4 sensors-23-01287-f004:**
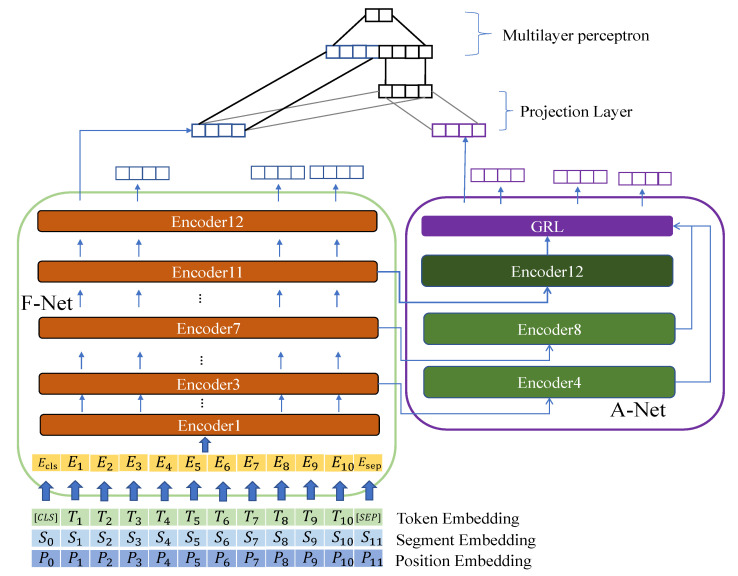
The figure shows the entire structure of the *FA-Net + BERT*. We implemented it based on our assumptions; thus, there are inputs for $E_{a}$ in the auxiliary network. In this figure, A-Net can extract different features according to different inputs of A-Net. For example, when the A-Net obtains the output of Encoder7, the backward features are extracted by Encoder8 rather than Encoder12. Additionally, the projection layer only works on the CLS token of two features. The encoder in dark green is the best encoder of $E_{a}$ that we tested in *FA-Net + RoBERTa* models.

### 3.2. Materials

To verify the effectiveness of our algorithm, we experimented with it by using six corpora, including a multilabel corpus and a binary-label corpora. The summarization of each corpus is shown in Table 1.

**MR** (https://www.cs.cornell.edu/people/pabo/movie-review-data/(accessed on 17 January 2023)): This corpus contains data on a document level, sentence level, sentiment scale, and subjectivity level. In our algorithm, we chose a sentence-level dataset to conduct the experiment on. It contains 4796 positive samples and 4796 negative samples.

**SST2** (https://nlp.stanford.edu/sentiment/(accessed on 17 January 2023)): This corpus contains 67350 positive and negative samples in the training dataset and 1821 samples in the testing dataset [20]. To conduct the experiment faster, we determined the difference between the benchmarks and our algorithm. We cut it down to 6920 training samples and 1821 testing samples.

**SemEval-2018 task 3** (https://github.com/Cyvhee/SemEval2018-Task3(accessed on 17 January 2023)): This task is named “Irony detection in English tweets.” [21] The task is part of the 12th workshop on semantic evaluation. It contains two labels: non-irony and irony. There are 1916 non-irony and 1901 irony samples in the downloaded training dataset and 472 non-irony and 310 irony samples in the downloaded test dataset.

**Sem-2017 task 4** (https://github.com/cbaziotis/datastories-semeval2017-task4(accessed on 17 January 2023)): This task is the subtask of SemEval-2017 [22]. Task 4 contains five tasks. This research only needs three tasks: A, C, and E. Task 4A contains 1188 negative, 2724 neutral, and 4088 positive samples. The label of task 4CE is the five-point scale for sets of tweets and topics. It contains 4159 *1point*, 2237 *0point*, 901 *-1point*, 585 *2point*, and 118 *-2point* sets.

**Waimai-10k** (https://gitee.com/sprite0153/ChineseNlpCorpus/tree/master/datasets (accessed on 17 January 2023)): This is a Chinese corpus. The data come from user reviews of a particular food delivery platform. It is a binary-label database containing 3612 positive examples and 7176 negative examples.

**Table 1 sensors-23-01287-t001:** The summarization of each corpus.

Corpus	Language	Labels	Samples of Training Data	Samples of Test Data
MR	English	2	7460	2132
SST2	English	2	6920	1820
SemEval-2018 task 3	English	2	3817	782
Sem-2017 task 4A	English	3	6000	2000
Sem-2017 task 4CE	English	5	6000	2000
Waimai-10k	Chinese	2	8390	2398

Note that some of those corpora have not separated data into training datasets. Therefore, we split the data into 80% training and 20% testing datasets if the original data was not split. We kept the split ratio if the original data had been split.

### 3.3. Experimental Benchmarks

To expressively verify the effectiveness of our model, we conducted experiments with three benchmarks to obtain the discrepant results. As BERT achieved state-of-the-art results for productive tasks, we utilized the vocabulary table of BERT to tokenize the original sentence at each benchmark to decrease the preprocessing time of the corpus.

TextCNN: As the extractor, we utilized the most representative model, CNN-rand, which is a TextCNN [1]. It uses a set of filters to capture the semantic feature maps and pool them. Then, it concatenates the features of different filter sizes to make the classification.

BERT: We utilized a pretrained BERT-based model, which includes 12 layers and 756 hidden sizes, to fine-tune the parameters of our datasets.

RoBERTa: The RoBERTa model also has huge pretrained models. Like BERT, we fine-tuned our corpus in the pretrained RoBERTa-based model.

Although we implemented three benchmarks, there is no difference in the settings between our algorithm and benchmarks, such that the batch size and convolution filters or other settings were not changed between the benchmarks and our FA-Nets model. Moreover, as we know, the initial seed greatly influences the model. We further considered the influence of seeds on the model, and thus, we calculated the average result under five seeds.

### 3.4. Experimental Settings

We fixed the embedding size to be 128-dimensional for each experiment, except for when using BERT and RoBERTa because the pretrained models cannot be changed. The setting details at each benchmark of our experiments are as follows:

FA-Net of TextCNN: The filter sizes are 3, and the n-grams are set to be (3, 4, and 5). The parameter of *L2-norm* is 0.001. The parameter of *Dropout* [23] is 0.5. In the A-Net, we empirically fixed the length of the filter sizes to 1 and set n-gram to be 4.

FA-Net of RoBERTa: Because our experiments contain four corpora, including English and Chinese datasets, we not only used the English pretrained model, which is called roberta-base (https://huggingface.co/roberta-base(accessed on 17 January 2023)), but we also chose the Chinese pretrained model, which is called hfl/chinese-roberta-wwm-ext (https://huggingface.co/hfl/chinese-roberta-wwm-ext(accessed on 17 January 2023)). However, we utilized the cardiffnlp/twitter-roberta-base-sentiment (https://huggingface.co/cardiffnlp/twitter-roberta-base-sentiment(accessed on 17 January 2023)) pretrained model for the sarcasm detection datasets.

FA-Net of BERT: Keeping it the same as RoBERTa, we utilized bert-base-uncased (https://huggingface.co/bert-base-uncased(accessed on 17 January 2023)) and bert-base-chinese (https://huggingface.co/bert-base-chinese(accessed on 17 January 2023)) pretrained models to conduct our experiments.

In the training stage, we fixed some hyperparameters corresponding to datasets at all benchmarks, such as batch size and the length of samples. The length of samples in all experiments was fixed to be 32 in waimai_10k, 64 in SST2, 64 in MR, and 256 in R8. In addition, the batch size was set to 32. For the other parameters in the BERT-based models, such as attention dropout, dropout, and weight decay, we kept the default setting that is applied by *Hugging Face*.

Because we utilized the different optimizers for the backpropagation stage and set different learning rates, except for the CNN-based models, the optimizer of the F-Net is Adam [24] with *β*_1_, *β*_2_ = 0.999. However, in the A-Net, the optimizer is the SGD optimizer where *moment* = 0.9. The difference between the CNN-based and BERT-based models and our FA-Nets have two optimizers and networks. Thus, the setting of the learning rate differs between models. Therefore, we set the learning rate to 0.001 in both optimizers in the FA-TextCNN models. However, in the FA-BERT-based models, we tried three kinds of learning rates (1e-5, 2e-5, and 3e-5) for the F-Net’s optimizer and set the learning rates to 0.001 for the optimizer of the A-Net.

## 4. Results and Discussion

### 4.1. Results

The evaluation indicator of the multicategory dataset is *F1-score*, and the evaluation indicator of the binary classification dataset is *accuracy* because all the datasets are classification corpora. The total parameters of the models are shown in Table 2 and the experiment results are shown in Table 3.

The models that start with “FA-” mean we added our auxiliary network to the original models. Additionally, we carried out different kinds of experiments to analyze the influences of different extractor sizes of the A-Net. The columns represent different datasets, and indices represent different algorithms.

**Table 3 sensors-23-01287-t003:** The contrast between benchmarks and our models is shown in this table. The results of BERT-based models are conducted under the five seeds, including benchmarks and FA-Nets. Task 3 means SemEval-2018 task 3, task 4A means Sem-2017 task 4A, and task 4CE means Sem-2017 task 4CE. In the TextCNN model, n_gram=n means the filter is set to n when initializing the A-Net’s convolution layer. In the BERT and RoBERTa models, *s* means the *s*th F-Net’s encoder is copied as the first encoder at A-Net, and *e* means the *e*th F-Net’s encoder is copied as the last encoder at A-Net. The input of A-Net is the output of the (s−1)th encoder of F-Net. Furthermore, the “OGRL” means that we removed the GRL in the A-Net. The *Avg* is the average result from the sentiment analysis and sarcasm detection datasets. The best results in our implementation are marked with bold font.

Model	SST2	MR	waimai_10k	Avg
TextCNN [1]	82.17 (±0.48)	76.68 (±0.27)	90.10 (±0.20)	82.98
BERT [2]	91.59 (±0.23)	86.38 (±0.22)	91.27 (±0.27)	89.74
RoBERTa [3]	94.43 (±0.31)	88.66 (±0.28)	88.73 (±0.34)	90.61
FA-TextCNN (n_gram = 3)	81.86 (±0.88)	76.68 (±0.56)	90.14 (±0.37)	82.89
FA-TextCNN (n_gram = 4)	**82.67** (±0.37)	**77.28** (±0.51)	**90.48** (±0.26)	**83.47**
FA-TextCNN (n_gram = 5)	82.56 (±0.62)	76.99 (±0.42)	90.32 (±0.21)	83.29
FA-BERT (s = 12, e = 12)	91.93 (±0.29)	86.91 (±0.45)	91.63 (±0.30)	90.16
FA-BERT (s = 11, e = 12)	91.79 (±0.35)	86.91 (±0.24)	91.39 (±0.23)	90.03
FA-BERT (s = 8, e = 8)	91.85 (±0.30)	87.14 (±0.22)	**91.69** (±0.20)	90.23
FA-BERT (s = 7, e = 8)	91.81 (±0.41)	86.88 (±0.21)	91.39 (±0.20)	90.03
FA-BERT (s = 5, e = 5)	**91.97** (±0.22)	**87.21** (±0.24)	91.60 (±0.16)	**90.26**
FA-BERT (s = 4, e = 5)	91.59 (±0.34)	86.82 (±0.33)	91.18 (±0.50)	89.98
FA-RoBERTa (s = 12, e = 12)	95.05 (±0.39)	**89.11** (±0.46)	**89.36** (±0.24)	**91.17**
FA-RoBERTa (s = 11, e = 12)	94.61 (±0.40)	88.91 (±0.37)	89.20 (±0.24)	90.91
FA-RoBERTa (s = 8, e = 8)	**95.11** (±0.32)	88.96 (±0.24)	89.25 (±0.17)	91.11
FA-RoBERTa (s = 7, e = 8)	94.62 (±0.15)	88.97 (±0.50)	89.23 (±0.24)	90.94
FA-RoBERTa (s = 5, e = 5)	95.03 (±0.34)	88.93 (±0.27)	89.35 (±0.29)	91.10
FA-RoBERTa (s = 4, e = 5)	94.82 (±0.14)	88.97 (±0.62)	89.34 (±0.09)	91.04
**Model**	**task 3**	**task 4A**	**task 4CE**	**Avg**
TextCNN	70.56 (±0.73)	45.04 (±1.07)	27.09 (±0.44)	47.56
BERT	71.17 (±1.58)	60.02 (±0.73)	39.99 (±0.90)	57.06
RoBERTa	72.70 (±0.71)	64.53 (±0.46)	43.50 (±1.62)	60.97
FA-TextCNN (n_gram = 4)	**71.62** (±0.87)	**46.39** (±0.73)	**27.85** (±0.89)	**48.62**
FA-TextCNN (n_gram = 4, OGRL)	71.32 (±1.19)	44.83 (±1.48)	26.07 (±1.45)	47.40
FA-BERT (s = 5, e = 5)	**72.42** (±1.06)	**60.95** (±0.52)	**41.62** (±0.65)	**58.33**
FA-BERT (s = 5, e = 5, OGRL)	71.48 (±0.79)	60.69 (±0.36)	41.03 (±0.56)	57.73
FA-RoBERTa (s = 12, e = 12)	**73.45** (±1.08)	**65.52** (±0.27)	**45.67** (±1.52)	**61.55**
FA-RoBERTa (s = 12, e = 12, OGRL)	72.16 (±0.93)	65.10 (±0.55)	44.44 (±1.24)	60.57

Following the different results in Table 3 for each line, we made some observations as follows:

In the sentiment analysis datasets, the *accuracy* of the TextCNN in three of the corpora is 82.17%, 76.68%, and 90.10%. However, the *accuracy* can become higher, reaching 82.67%, 77.28%, and 90.48%, and the average *accuracy* under those corpora received a 0.5% boost when the model added our auxiliary network. This shows that our auxiliary network can boost the original models by adding acceptable parameters in the TextCNN. The same results occurred in BERT and RoBERTa. The average *accuracy* of BERT and RoBERTa can also receive a 0.5% boost due to our auxiliary network. The *accuracy* can receive a 0.8% boost for MR and SST2 when our auxiliary network is added to BERT and RoBERTa.

We utilized the best hyperparameters from the experiments on the sentiment analysis datasets to implement our methods on the sarcasm detection datasets. The effectiveness of our proposed method is higher than that of the sentiment analysis. Compared with the TextCNN, the FA-TextCNN models have an average improvement of 1% for three datasets. Compared with BERT, FA-BERT has an average improvement of 1.3% in the SemEval-2018 task 3, 0.9% in the Sem-2017 task 4A, and 1.6% in the Sem-2017 task 4CE. Compared with RoBERTa, FA-RoBERTa has an average improvement of 0.8% in the SemEval-2018 task 3, 1% in the Sem-2017 task 4A, and 2.1% in the Sem-2017 task 4CE. Even if we removed the GRL, our proposed algorithms still cause an improvement, but the magnitude of the improvement is lower than in the models that are stacked with the GRL. The difference that boosts the effectiveness between the sentiment analysis and sarcasm detection datasets is caused by the proposed method, since its core idea is to scatter the semantic information of contradictory words to other words in a sentence. Additionally, the sarcasm detection datasets have many of the sentences that contain contradictory words. Therefore, the boosted effectiveness of our proposed method on the sarcasm detection datasets is higher than on the sentiment analysis datasets. As seen in Table 3, the results show that even in different settings, our algorithm still can improve the benchmarks’ performance. In the CNN-based model, our FA-TextCNN is more stable than the TextCNN when the A-Net chooses a filter that fixed the n-gram = 4 as an extractor. However, compared to the other settings, the algorithms are not stable. Our algorithms are worse than CNN benchmarks when the n-gram = 3. We assumed that unstable conditions are due to the *dropout* because the corresponding deviation of the CNN benchmark is close to 0.5% in the SST2 and MR datasets. This means that the original CNN model is unstable, as it is simply uninterpretable. Therefore, when we fused the projected features that were extracted from normal features, the property further influences the stability of our FA-TextCNN models.

The results echo our previous assumption that semantic information can still be exploited in the backward features at high-level layers in BERT-based and RoBERTa-based models. After conducting the different experiments, we also studied the influence of the FA-Net on the attention heat map, which is explained in the Discussion.

### 4.2. Discussion

We studied the influence of the number of encoders and the projection types on the FA-Net. As the CNN is poor at extracting high-quality features, fusing the backward features directly through Equation (Equation 10) can hurt the performance of the CNN model. Thus, this discussion omits the CNN models. Both further experiments are focused on the BERT-based and RoBERTa-based models. Equation (Equation 10) projects $v_{a}$ to $v_{f}$. In other words, the result of Equation (Equation 10) acquires the subfeature of $v_{f}$. From this point, we can discuss another projection type, which projects $v_{f}$ to $v_{a}$, to further study which projection type is better for our idea.

**Projection type**: Figure 5 shows the experiments that change the projection type to Equation (Equation 11). According to Table 3, the best results appear in different types of the A-Net. In the *FA-Net + BERT* model, the most effective focuses on the encoder in the middle, but in *FA-Net + RoBERTa*, the most effective is the encoder at the top. Thus, we implemented more experiments by using Equation (Equation 11) for SST2 with different encoders.

As seen in Figure 5, there is improvement when the first encoder of the A-Net is between four and six, but the improvements are minor for Equation (Equation 10). Additionally, the accuracy becomes lower when the A-Net’s first encoder is copied from the top of the F-Net. This means that Equation (Equation 11) is too unstable to conflict with our idea. We assumed that the gradient of $v_{a}$ can influence several encoders of the F-Net when *x* is larger than nine and then further influence the model’s performance even if we fuse a part of $v_{a}$.

**Figure 5 sensors-23-01287-f005:**
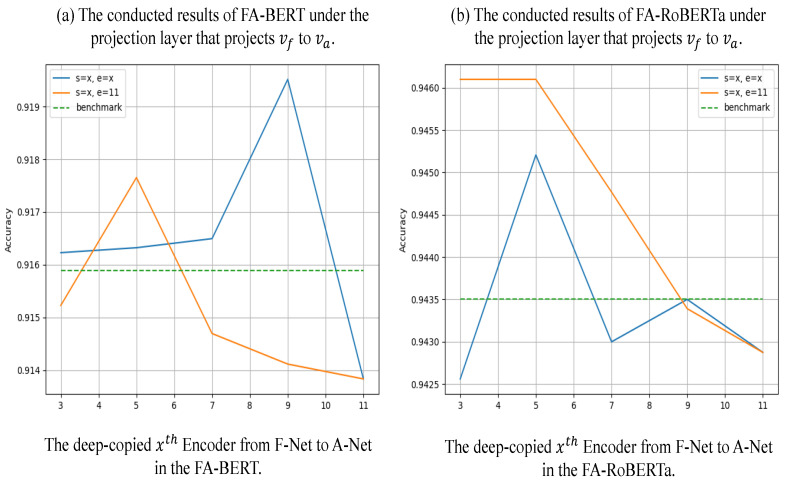
In this figure, we concatenate $v_{f}$ with ${\bar{v}}_{a}$ to implement our FA-Nets. Both models are implemented in the SST2 dataset. Chart *a* is the BERT-based model, and chart *b* is the RoBERTa-based model. All results shown in this figure are averaged from the five seeds. The **blue** line is labeled by *s = x, e = x*, which means the first encoder and the last encoder in A-Net are copied from the $x^{th}$ encoder in F-Net. Furthermore, the **orange** line means the last encoder of A-Net is fixed to *11*, but the first encoder is changed with *x*. Moreover, the **dotted green** line results from corresponding benchmark models.

However, if we feed the outputs of the three to nine layers to the A-Net, our network’s accuracy is also competitive. We discovered that the influence is lower when feeding low-level outputs to the A-Net because the gradient of $v_{a}$ only affects a few layers of the low-level encoders. The influence becomes higher as the A-Net obtains the high-level outputs. Because of the gradient of $v_{a}$, the feedback to the F-Net is early. According to the projection type analysis and discussion, we empirically fused a part of $v_{f}$ to concatenate more semantic information rather than fuse part of $v_{a}$ in order to achieve our ideas. We assumed that the normal vector $v_{f}$ is augmented by concatenating with ${\bar{v}}_{f}$, as it contains the essential abstract information relative to high-gradient features. Concatenating with ${\bar{v}}_{f}$ also can avoid the A-Net feedback that reverses the gradient of $v_{a}$ to the F-Net.

**The number of encoders of the A-Net**: The BERT and RoBERTa models are perplexing; thus, the BERT-based model needs to consider how many encoders should be copied to the A-Net. According to Table 3, the best results occurred when the A-Net had one encoder. Furthermore, we can obtain average results for MR, SST2, and waimai_10k, as shown in Table 3. The average results of the multi-encoder that exist in the A-Net are lower than using a single encoder in the A-Net. This shows that the gradient reversal layer with a multi-encoder can learn a higher gradient feature than a single encoder. Therefore, a multi-encoder is more unsuitable than the single encoder in these corpora.

**The attention heat map of an example sentence in FA-RoBERTa**: We tested the sample we mentioned previously and drew an attention heat map. Compared to Figure 6, in Figure 7, although attention is paid to the phrase “ultimately silly”, it does not focus on just this phrase anymore. FA-RoBERTa focuses on the core phrase, “charming and funny”. Then, FA-RoBERTa makes the right classification. This proves that our idea could scatter the attention score of contradictory words to other words to reduce the destructive influence of contradictory words. As long as the model spreads the attention score to other words, the semantic information of other words can fuse with the normal features after concatenating. Thus, the classifier can capture more information about other words rather than focusing on the contradictory words.

**Figure 6 sensors-23-01287-f006:**
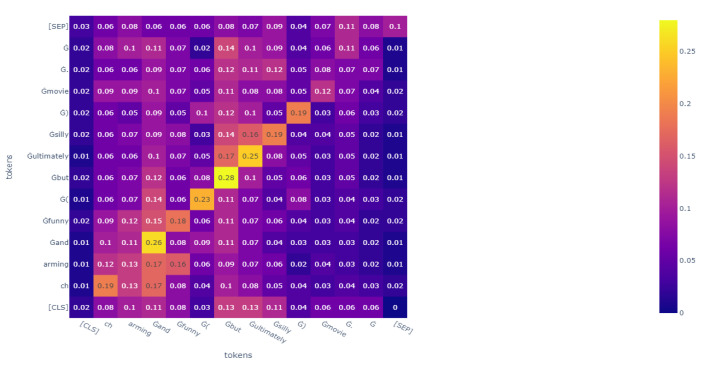
The attention heat map of phrase, “charming and funny ( but ultimately silly ) movie.”, in the RoBERTa model.

## 5. Conclusions

In this paper, we proposed a concise two-stream network that combines the extracted feature vectors to absorb more semantic information, allowing a sentence that contains special phrases which are opposite to the semantic information of the whole sentence to be classified correctly. In addition, we utilized the feature projection layer, gradient reversal layer, and vector concatenate to achieve this goal. Based on the original model, we considered the size of the parameters and the accuracy of the model, and we developed a proper model to avoid huge parameters and low accuracy. To prove the effectiveness of our algorithm, we conducted contrast experiments on sentiment analysis and sarcasm detection datasets. Ultimately, the results show that our algorithms are effective for those benchmarks. After implementing extra experiments, we further studied the influence of projection type and the number of encoders of the A-Net. We determined the general hyperparameters for three benchmarks in several datasets.

The current algorithm focuses on text classification. In the future, we will examine other NLP tasks using the short- and long-length sentence corpus or document corpus, and we will try to study the impact of data granularity in meta-learning.

## Figures and Tables

**Figure 1 sensors-23-01287-f001:**
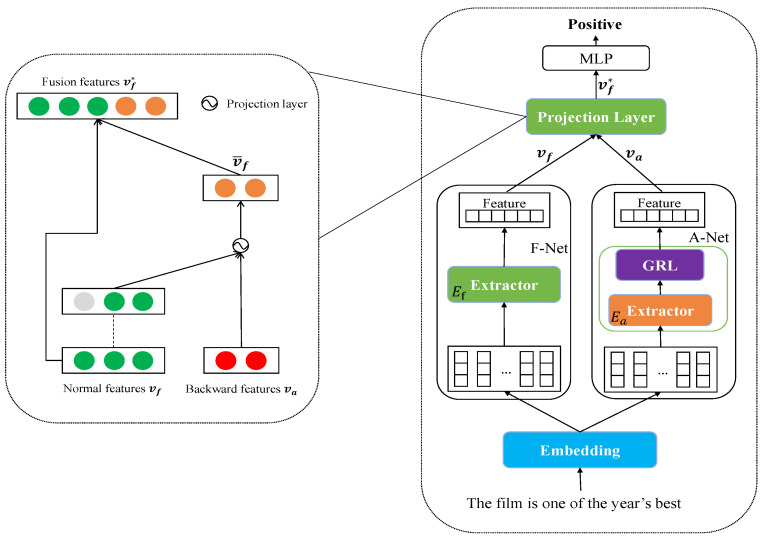
The architecture of FA-Net. The right structure shows the detail of the projection layer. The left one displays the whole network as part of the right boxed figure. Each component is the same except for the extractor of F-Net and A-Net. Note that the “Embedding” concludes with the blue box, but it is not the embedding layer. Rather, it is an abstract layer. The outputs of “Embedding” may be produced by the embedding layer or by one of the transformer encoder layers.

**Figure 2 sensors-23-01287-f002:**
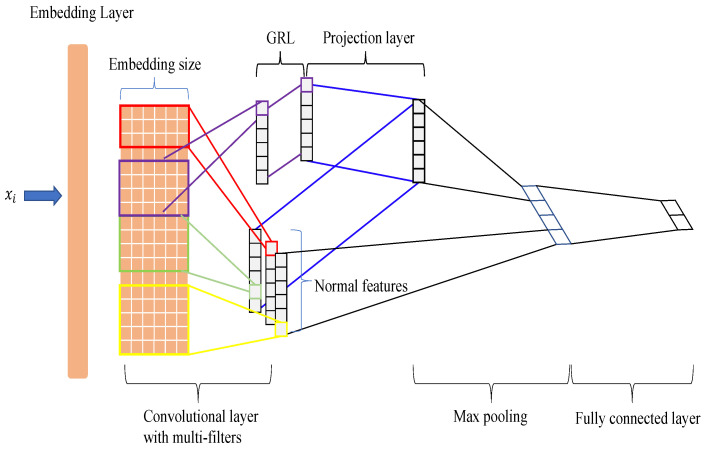
The total structure of the *FA-TextCNN*. The purple line is the extractor of the auxiliary network that utilizes a gradient reversal layer. As shown in the figure, we concatenated the features after maxpooling and fed the features together to form a fully connected layer.

**Figure 3 sensors-23-01287-f003:**
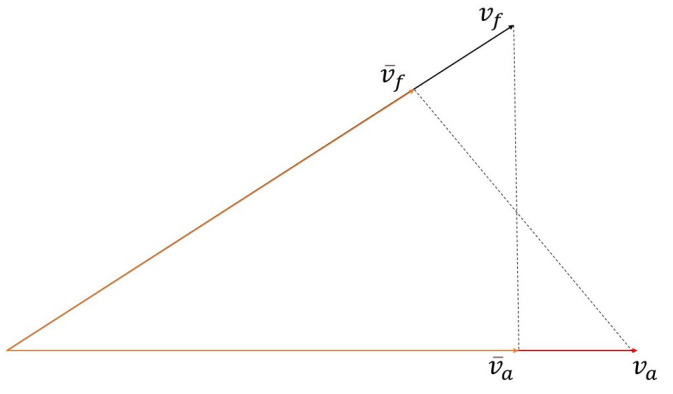
Two-dimensional progress of projection function. $v_{a}$ is the feature vector extracted by A-Net, and $v_{f}$ is the feature vector extracted by F-Net. ${\bar{v}}_{f}$ is the new feature vector that $v_{f}$ projected to $v_{a}$, and ${\bar{v}}_{a}$ is the new feature vector that $v_{a}$ projected to $v_{f}$.

**Figure 7 sensors-23-01287-f007:**
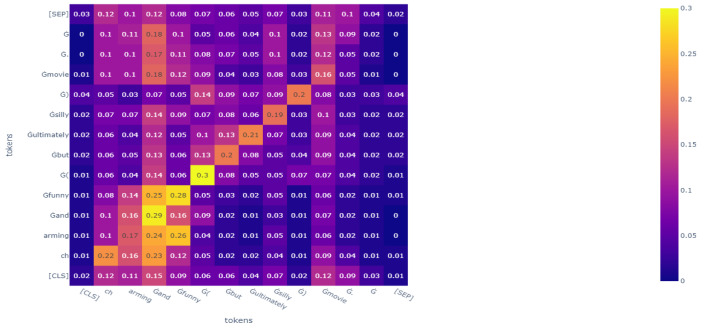
The attention heat map of phrase, “charming and funny ( but ultimately silly ) movie.”, in our FA-RoBERTa model.

**Table 2 sensors-23-01287-t002:** Comparison of total parameters between base models and our models: +1En means the auxiliary network has one transformer encoder, and +2En means two transformer encoders exist in the auxiliary network.

	TextCNN	FA-Net + TextCNN	
#param	40M	41M	
	**RoBERTa**	**FA-Net + RoBERTa**	
#param	125M	134M(+1En)	141M(+2En)
	**BERT**	**FA-Net + BERT**	
#param	109M	119M(+1En)	126M(+2En)

## Data Availability

Not applicable.

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
