# Peer review of "Enhancing the Generalization for Text Classification through Fusion of Backward Features"

_sensors, 2023, doi:10.3390/s23031287_

Round 1

Reviewer 1 Report (Previous Reviewer 2)

This paper shows an interesting framework to improve the generalization ability of text classification models, which is a very relevant and important topic. However, the paper still has a huge number of English errors that make it difficult to comprehend some parts of the manuscript. I recommend seeking professional help to improve the language of the paper before resubmitting it.

Author Response

Replies to Comments from Reviewer #1

Comment 1: “However, the paper still has a huge number of English errors that make it difficult to comprehend some parts of the manuscript. I recommend seeking professional help to improve the language of the paper before resubmitting it.”

Response: Thanks to reviewer#1 for the unbiased comments, the writing issues of our manuscript are indeed very problematic. We have used the editing services provided by your journal for this purpose. When we got the grammatically corrected manuscript, we found that many grammatical errors had been corrected accurately. Based on the revised manuscript, we used the kdiff software to track the revised words and highlighted most of the revised terms, and the highlighted parts have been displayed in the revised manuscript.

Reviewer 2 Report (New Reviewer)

This manuscript describes an auxiliary network to improve the representativeness of normal features and the generalization ability of the main network. There is a high interest in the study of the generalization ability. But there are some problems I don't understand.

1. It is suggested to add some introduction to the method, such as the GRL.

2.It is suggested to add first line indent in some paragraphs, such as the page5.

3.It is suggested to add a graphical abstract.

Author Response

Replies to Comments from Reviewer #2

Comment 1: “It is suggested to add some introduction to the method, such as the “GRL.”

Response: After reading, we both think more than the explanation of GRL is needed. Based on this issue, we replaced the GRL with the Gradient reversal layer in the abstract and the main text and explained the role of GRL in the abstract. At the same time, we explained why inverting the gradient can reduce the semantic information of contradictory words on page 4. On the sixth page, we exchanged the explanation of GRL and Ea, put the description of Ea generation features in front, and put GRL. The explanation of the gradient is placed later to illustrate the order of Ea and GRL. Finally, we also explained at the end of page 13 that after normal features are added to reversed gradient features, the attention of other words can be improved, and the model's attention can be refocused on the core words. Both revisions can be found in the revised manuscript.

Comment 2: "It is suggested to add first line indent in some paragraphs, such as the page5.”

Response: This problem was indeed due to our oversight. We checked all indents in the manuscript and corrected inappropriate ones.

Comment 3: "It is suggested to add a graphical abstract."

Response: This is a very valuable suggestion, and the graphical abstract can intuitively illustrate our method, allowing readers to understand the significance of our research at first sight of the picture. So we added a picture to abstractly show our work and explain the research process from the character level.

Reviewer 3 Report (New Reviewer)

This work describes a method to scatter the attention of opposite sentiment words to others to avoid one-side judgment.

The work reflects a novel approach on facing this problem but requires further experimentation with more expressions to conclude that it works properly. However, I suggest the acceptance of this paper once this is explained as a limitation.

Some minor English issues should also be addressed, such as:

- Remove the point in this sentence: "The most classic model among those neural network models is Convolutional Neural Network.[1]".

- Remove the s in this sentence: "capture the corpus’s universal information".

- Use "Pre-trained technology" instead of "pre-train technologies".

- Or "To expressively verify the effectiveness of our model, we conduct experiments", should be "conducted".

Author Response

Replies to Comments from Reviewer #3

Comment 1: The work reflects a novel approach on facing this problem but requires further experimentation with more expressions to conclude that it works properly.

Response: This is a very pertinent piece of advice. We considered the applicability of our proposed method in subsequent revisions, and we assume that the method in the paper is more suitable for use in the task of sarcasm detection. Therefore, based on the results of experiments in the sentiment analysis dataset, we further conducted experiments on the sarcasm detection dataset. Of course, the sarcasm detection datasets are all open-source datasets. After comparing the experimental results on the sentiment analysis dataset and the sarcasm detection dataset, we reasonably explained why our method has different improvement effects under the two datasets. In addition, we have also modified some of the picture errors, the purpose of which is to make the content of the picture more in line with our proposed method.

Comment 2: Some minor English issues should also be addressed, such as: Remove the point in this sentence: "The most classic model among those neural network models is Convolutional Neural Network.[1]". Remove the s in this sentence: "capture the corpus’s universal information". Use "Pre-trained technology" instead of "pre-train technologies". Or "To expressively verify the effectiveness of our model, we conduct experiments", should be "conducted".

Response: We would like to thank reviewer 3 for his careful and responsible review. For this reason, we have revised all the errors raised, and we also used the polishing service provided by the journal to help us correct more grammatical and formatting errors. In the revised manuscript, we have highlighted all revisions.

Round 2

Reviewer 1 Report (Previous Reviewer 2)

The manuscript has been improved significantly. The Topic and Experiment is very interesting and relevant. Therefore, I recommend the acceptance of the paper.

This manuscript is a resubmission of an earlier submission. The following is a list of the peer review reports and author responses from that submission.

Round 1

Reviewer 1 Report

The paper presents a new model for text classification using a two-stream network and stack the gradient reversal layer and the feature projection layer in an auxiliary network. The effectiveness of the method is shown by comparing it with three other models: TextCNN, BERT, and RoBERTa.

The paper is incomprehensible and the results described in Table 3 do not show that the two-stream network perform better than the other models. So, the paper should be rejected.

BERT (Bidirectional Encoder Representations from Transformers) is referred to as Bert. This is wrong.

There are many sentences in the paper that do not make sense:

Lines 21-22: There has heavy applications in this field, such as news classification [1], natural language inference [2], sentiment classification [3].

Lines 25-26: The effects of many NLP tasks have been greatly improved with the help of neural networks.

Lines 138-139: They construct a graph that calculates the distance between documents with consine.

Lines 146-147: Both of us do not need to introduce additional data. But different from GhostBert.

The results explained on Page 12 do not make any sense:

Even in the RoBERTa model, we also got a 0.45% 279

boost. The improvement effect on the SST2 is not apparent as MR in both models. An 280

improvement of 0.5% in the CNN model, 0.68% in the RoBERTa model, and 0.44% in the 281

Bert model. The effect on waimai_10k is not apparent in CNN and Bert models; it just has 282

an improvement of 0.38% in the CNN model and 0.41% in the Bert model. But there is an 283

0.63% improvement in the RoBERTa model. Last, the less efficient dataset is R8 in both 284

models. All the improvements are less than 0.4%. We thought the character of the dataset 285

caused this problem.

Reviewer 2 Report

This paper proposes a framework to improve the generalization ability of text classification models. This is an interesting topic. However, the paper has many flaws that need to be addressed:

- There are several grammatical errors that need to be corrected, for example in sentences 5 and 6:

"those models are unstable when classifying the sentence, which label is positive(negative) but contain negative(positive) phrases"

There are tens of similar errors that make it difficult to understand the manuscript.

- Please avoid using unexplained acronyms in the abstract.

- Bert should be BERT; this is an acronym that should be introduced and used as such.

- It is unusual to have a figure in the introduction.

- The caption for figure 1 is taken from the introduction text; please use a succinct phrase.

- Line 75: "The structural detail of the two networks will describe in Section3" the manuscript has tens of errors like this one

- Line 76: says that there are 4 contributions, but only 3 were mentioned.

- Introduction should end with a paragraph describing the organization of the rest of the manuscript.

- Line 86: "But this model will occur the problem named vanishing gradient." Many sentences like these occur throughout the manuscript, which limits my ability to comprehend what I am reading.

- Line 154: "The overview of the struct of our FA-Net is described in Figure 1." Do you mean figure 2, because figure one shows a heatmap.

- Line 219: "Note that some of those corpora have not separated data into training datasets and testing datasets, so we split the data according to about 80% of the training datasets, 20% of the testing datasets, and 10% of the validation datasets." This is vague. Which corpora are you talking about? And why did you choose this split ratio? And what do you mean by "about 80% of the training dataset"?

- What is the C column in table 1?

- Could you explain why you chose accuracy as an "evaluation standard"?

- Could you please explain the significance of the enhancements mentioned in the first paragraph of page 12?

- Line 285: "We thought the character of the dataset caused this problem." What do you mean by this?

- References need to be updated; most of the references are old.